# Technical Note: The CREDIBLE Uncertainty Estimation (CURE) toolbox: facilitating the communication of epistemic uncertainty

Trevor Page[1], Paul Smith[1,2], Keith Beven[1], Francesca Pianosi[3], Fanny Sarrazin[4], Susana Almeida[5], Liz Holcombe[3], Jim Freer[6], Nick Chappell[1], and Thorsten Wagener[7]

[1] Lancaster Environment Centre, Lancaster University, Lancaster, UK

[2] Waternumbers, Lancaster, UK

[3] Department of Civil Engineering, Bristol University, Bristol, UK

[4] Department of Computational Hydrosystems, Helmholtz Centre for Environmental Research (UFZ), Leipzig, Germany

[5] Atkins Global, Warrington, UK

[6] School of Geographical Sciences, Bristol University, UK and Global Institute for Water Security, University of Saskatchewan, Canada

[7] Institute for Environmental Science and Geography, University of Potsdam, Germany

*Correspondence to*: Keith Beven (k.beven@lancaster.ac.uk)

**Abstract.** There is a general trend for increasing inclusion of uncertainty estimation in the environmental modelling domain. We present the CREDIBLE Uncertainty Estimation (CURE) Toolbox, an open source MATLAB[TM] toolbox for uncertainty estimation aimed at scientists and practitioners that are not necessarily experts in uncertainty estimation. The toolbox focusses on environmental simulation models and hence employs a range of different Monte Carlo methods for *forward* and *conditioned* uncertainty estimation. The methods included span both formal statistical and informal approaches, which are demonstrated using a range of modelling applications set up as *workflow scripts.* The workflow scripts provide examples of how to utilise toolbox functions for a variety of modelling applications and hence aid the user in defining their own workflow: additional help is provided by extensively commented code. The toolbox implementation aims to increase the uptake of uncertainty estimation methods within a framework designed to be open and explicit, in a way that tries to represent best practice in applying the methods included. Best practice in the evaluation of modelling assumptions and choices, specifically including *epistemic* uncertainties, is also included by the incorporation of a *condition tree* that allows users to record assumptions and choices made as an *audit trail log*.

## 1 Introduction

Environmental simulation models are used extensively for research and environmental management. There is a general trend for increasing inclusion of uncertainty estimation (UE) in the environmental modelling domain,

including applications used in decision making (Alexandrov *et al*., 2011; Ascough *et al*., 2008). Effective use of model estimates in decision making requires a level of confidence to be established (Bennett *et al*., 2013) and UE is one element of determining this. Another required element is an assessment of the *conditionality* of any UE; i.e. the conditionality associated with the implicit and explicit choices and assumptions made during the modelling and UE process, given the information available (e.g. see Rougier and Beven, 2013).


Here we present the CREDIBLE Uncertainty Estimation (CURE) Toolbox; an open source MATLAB$^{TM}$ toolbox for UE associated with environmental *simulation* models. It is aimed at scientists and practitioners with some modelling experience, that are not necessarily experts in UE. The toolbox structure is similar to that of the SAFE toolbox (Sensitivity Analysis For Everyone; Pianosi *et al*., 2016) such that it allows more experienced users to

modify and enhance the code and to add new UE methods. The implementation of the toolbox also aims to increase the uptake of UE methods within a framework designed to be open and explicit, in a way that tries to represent best practice. That is, best practice in applying the various UE methods included as well as best practice in being explicit about modelling choices and assumptions.

As the focus of the toolbox is UE for simulation models, often with relatively complex structures and many model parameters, the toolbox employs a range of different Monte Carlo methods. These are used for forward propagation of uncertainties by sampling from *a priori* defined input and parameter distributions, for *forward UE,* or in the estimation of refined model structures and/or associated posterior parameter distributions when conditioned on observations (*conditioned UE*). The methods included span both formal statistical and informal approaches to UE,

which are demonstrated using a range of modelling applications set up as *workflow scripts* that provide examples of how to utilise toolbox functions. As noted in the comments in the code, many of the Workflows can be linked to the description of methods in Beven (2009).

Formal statistical and informal methods are included as there are no commonly agreed techniques for UE in

environmental modelling applications, as evidenced by continuing debates and disputes in the literature (e.g. Clark *et al*., 2011; Beven *et al*., 2012; Beven, 2015; Nearing *et al*., 2016). The lack of consensus on the most appropriate UE method is to be expected given that the sources of uncertainty associated with environmental modelling applications are dominated by a lack of knowledge (*epistemic* uncertainties; e.g. see Refsgaard *et al*., 2007, Beven, 2009; Beven *et al*. 2016; Beven and Lane, 2022) rather than solely random variability (*aleatory* uncertainties).

Rigorous statistical inference applies to the latter, but might lead to unwarranted confidence if applied to the former,

especially where some data might be disinformative in model evaluation (e.g. see Beven and Westerberg, 2011; Beven and Smith, 2015; Beven, 2019; Beven and Lane, 2022).

Assessing the impact of epistemic uncertainties for environmental modelling requires assumptions about their nature (which are difficult to define) such that the outputs from any UE will be conditional upon these assumptions. This poses the question of what is good practice in evaluating assumptions and choices made during the modelling process and what is good practice in communicating the meaning of any subsequent analyses (Walker *et al*., 2003; Sutherland *et al*., 2013; Beven et al., 2018b; see also the TRACE framework of Grimm *et al*., (2014) for documenting the modelling process). Beven and Alcock (2012) suggest a *condition tree* approach, that records the

modelling choices and assumptions made during analyses and thus provides a clear *audit trail* (e.g. see Beven *et al*., 2014; Beven and Lane, 2022). The audit trail consequently provides a vehicle that promotes transparency, best practice and communication with stakeholders (Refsgaard *et al*., 2007; Beven and Alcock, 2012). To encourage best practice, the process of defining a condition tree and recording an audit trail has been made an integral part of the CURE Toolbox via a condition tree Graphical User Interface (GUI).


Other freely available toolboxes include for *forward UE**:* the Data Uncertainty Engine[1] (DUE; Brown *et al*., 2007) and the SIMLAB[2] toolbox (Saltelli *et al*., 2004); and for *conditioned UE**:* GLUEWIN [3] (Ratto and Saltelli, 2001), UCODE 2014[4] (Poeter *et al*., 2014), the Monte Carlo Analysis Toolbox[5] (MCAT; Wagener and Kollat, 2007), the MATLAB UQLAB[6], the Interactive Probabilistic Prediction software[7] (McInerney et al., 2018) and the DREAM[8]

Toolbox (Vrugt *et al*., 2008; 2009; Vrugt, 2016). See also the broader review of uncertainty tools undertaken by UNCERTWEB (Bastin *et al*., 2013) which includes tools supporting elicitation, visualisation, uncertainty and sensitivity analysis. While links exist for these toolboxes, it is not clear that all continue to be maintained and supported. The CURE Toolbox presented here is open source and brings together formal and informal modelling methodologies, underpinned by different philosophies, that users are encouraged to explore via the example

workflows (Table 1). It also includes a method not included in previous toolboxes (i.e. the GLUE-Limits of

---

[1]*http://harmonirib.geus.info/due_download/index.html;*
[2]*https://ec.europa.eu/jrc/en/samo/simlab;*
[3]*https://joint-research-centre.ec.europa.eu/macro-econometric-and-statistical-software/econometric-software/gluewin_en;*
[4]*http://igwmc.mines.edu/freeware/ucode;*
[5]*http://www.imperial.ac.uk/environmental-and-water-resource-engineering/research/software/;*
[6]*http://www.uqlab.com;*
[7]http://www.probabilisticpredictions.org;
[8] *http://faculty.sites.uci.edu/jasper/files/2015/03/manual_DREAM.pdf*

Acceptability (LoA) method; Beven, 2006; Blazkova and Beven, 2009; Hollaway *et al.,* 2018; Beven et al., 2022) and explicitly sets out to encourage best practice regarding the conditionality of modelling results using the condition tree approach.

## 2 Choosing a Workflow

Table 1 lists the example Workflows included in the first release of the CURE Toolbox and the methods employed, with references to published papers where the methods have been applied.   A variety of Workflows covering both Forward UE and both formal statistical and informal methods of Conditioned UE.  Figure 1 provides an illustration of the choices that might be made in deciding on a Workflow within the CURE Toolbox (see also the earlier

decision trees of this type in Pappenberger et al., 2006; Beven, 2009).   Forward UE methods (Workflows 1, 2) must be used when there are no observational data with which to condition the model outputs.  The outcomes will then be directly dependent on the assumptions about prior distributions and covariation of parameters and input variables.   Copula methods are used to sample covariates (Workflows 3,4).    In the case of both Forward and Conditioned UE Workflows, input uncertainties are parameterised to be applied as ranges or distributions, for

example, as multipliers or an additive bias applied when the model is run.

When observational data are available, formal statistical likelihood methods (Workflows 5 and 6) will be most appropriate in cases where any model residuals can be assumed to be aleatory and represented by a simple stochastic model.   Where such assumptions are difficult to justify because of epistemic sources of uncertainty, then there is

a choice between Approximate Bayesian Estimation (ABC) using Monte Carlo Markov Chain (MCMC) sampling and GLUE methods.       Within ABC, a threshold of acceptability for some informal summary measure of performance is chosen.  The MCMC sampling is implemented using the DREAM code described in Vrugt (2015; see Vrugt, 2016, for a more recent description).  This aims to produce an ensemble of model parameter sets comprising the samples from the final iterations of the DREAM algorithm (defined by the user) that are considered

as equally probable (Workflows 7,8).  Convergence of the sampling can be tested using the Gelman and Rubin (1992) diagnostic statistic.

Within GLUE each model is associated with a likelihood measure that initially reflects sampling of the assumed prior distributions and is then modified during the conditioning process.  GLUE allows for different ways of

updating the likelihood measure including both Bayesian multiplication and fuzzy operators (Beven and Binley,

1992, 2014). Uniform independent priors across specified ranges are often assumed when there is a lack of robust knowledge about the parameters but, as in the options for the forward UE workflows, other prior distributions can be used. Deciding on whether a model is acceptable or behavioural can again be based on some informal summary measure of performance (Workflows 9,10) or some predefined limits of acceptability (Workflows, 11,12). A

particular case of defining limits of acceptability for rainfall-runoff models based on historic event runoff coefficients as a way of reflecting epistemic uncertainties in observed inputs and outputs is included (Workflows 13,14). Vrugt and Beven (2018) have demonstrated an adaptive sampling methodology for applying the limits of acceptability (DREAM$_{(LoA)}$) that aims to find feasible samples that satisfy all the limits applied. The DREAM algorithm used in Workflows 7 and 8 can be adapted to be used in this way.


It should be noted that the examples associated with each workflow are intended to be illustrative. They cannot all be described in detail in this Technical Note intended to introduce the Toolbox. However, the Matlab$^{TM}$ code is freely available and can be easily adapted by users for their own applications. Extensive comments are included in each workflow to aid this process.


*Table 1 – Toolbox workflow examples and uncertainty estimation methods employed*

| Workflow | Uncertainty estimation method | Example Model | Shot Description |
|---|---|---|---|
| 1 | Forward (independent sampling) | CHASM[1,§] | Application of uniform sampling of statistical distributions |
| 2 | Forward (independent) | HYMOD[2] | Fuzzy parameter distributions |
| 3 | Forward (dependent sampling - copula) | HYMOD | Marginal gamma distributions and rank correlation derived copula |
| 4 | Forward$^{€}$ (dependent sampling - copula) | LISFLOOD[3, §] | Application of covariant model[4] fitted as copula for inflows to LISFLOOD |
| 5 | Conditioned; Adaptive Metropolis MCMC[4,5,6] | HYMOD | Single chain MCMC; formal likelihood |
| 6 | Conditioned; DREAM[7] | HYMOD | Multi-chain MCMC; formal likelihood |

| 7 | Conditioned; DREAM [8] [ψ] | HYMOD | Multi-chain MCMC with thresholding of informal likelihood measure |
|---|---|---|---|
| 8 | Conditioned; DREAM | PROTECH[4, §] | Multi-chain MCMC using thresholding of informal likelihood measure |
| 9 | Conditioned; GLUE[9] | HYMOD | GLUE using threshold of informal likelihood measure |
| 10 | Conditioned; GLUE | PROTECH[§] | GLUE using threshold of informal likelihood measure. |
| 11 | Conditioned; GLUE-LoA | HYMOD | GLUE using single-variable Limits of Acceptabilty |
| 12 | Conditioned; GLUE-LoA[10] | PROTECH[§] | GLUE using multi-variable Limits of Acceptability |
| 13 | Analysis of rainfall-runoff observations | Event analysis[11] | Derivation of Limits of Acceptability based on event runoff coefficients |
| 14 | Conditioned GLUE-LoA | Dynamic TOPMODEL[11] | GLUE using Limits of Acceptability based on runoff coefficients |

[€] *In this example the inputs were sampled in a forward uncertainty analysis but the LISFLOOD model was conditioned in a prior analysis;* [1] *Almeida et al. (2017);* [2] *Wagener et al. (2001);* [3] *Neal et al. (2013);* [4]*Haario, et al. (2001);* [5] *Roberts and Rosenthal (2001);* [6] *Roberts and Rosenthal (2009);* [7] *DiffeRential Evolution Adaptive Metropolis, Vrugt (2016);* [8] *Sadegh and Vrugt (2014);* [9] *Generalised Likelihood Uncertainty Estimation, Beven and Binley (1992);* [10] *Blazkova and Beven (2009);* [§] *Owing to long model run times this example uses pre-run simulation output;*[11] *Beven et al. (2022)* [ψ] *Approximate Bayesian Computation;* [¥]*Limits of Acceptability.*

## 3 The CURE Toolbox Version 1.0 Structure

The CURE Toolbox essentially has two linked structures. There is an overall structure with which the user interacts throughout the analysis (Figure 2) and an underlying folder structure (Figure 3) containing the toolbox functions and example model-specific files. The toolbox folder structure has specific folders for the UE methods where method-specific functions are collated (e.g. method-specific sampling, diagnostics and visualisation) and for the

individual example modelling applications (i.e. the model functions and input files as well as any links to any *'external models'*: i.e. models not coded as a MATLAB<sup>TM</sup> function, but which can be executed from the command line). Folders also exist for general (i.e. not method-specific) sampling methods, visualisations and utility functions. Additionally, there are project folders for each example workflow where *audit trail logs*, diagnostics and results are written.

The functions for general sampling of parameter distributions (e.g. uniform, low discrepancy or Latin Hypercube sampling of the large number of supported distributions) are common with the SAFE toolbox of Pianosi et al. (2016). In addition, and of particular importance for forward uncertainty analysis, the sampling functions have been extended to represent parameter and forcing-input dependencies using copulas (e.g. Workflow 3 in Table 1 uses copula sampling based on results from previous analyses to describe parameter dependencies for forward uncertainty propagation). Other specific sampling functions are associated with the adaptive sampling ("*on-line*" sampling) for Markov Chain Monte Carlo (MCMC) approaches, implemented using the DREAM algorithm of Vrugt (2016), where distributions and correlation structures are modified as the chain(s) evolve. Modelling diagnostics, both numeric and graphical are provided for both on-line adaptive sampling and "*off-line*" methods (i.e. those that are not adaptively sampled within a given method). In the case of on-line MCMC methods, visualisation of the evolution of the states of the chain(s) and tests for convergence to stationary distributions are included (e.g. Figure 4 a and b).

In the case of formal statistical likelihood methods (see, for example, Evin et al., 2013, 2014, and the recent "universal likelihood" of Vrugt et al., 2022), residual model fitting can be carried interactively, using command line prompts, and can form part of a workflow (or used stand-alone). The approach uses Box-Cox transformations which provide flexibility in transforming the data to remove heteroscedasticity and non-normality (Box and Cox, 1964), and also provides for fitting an autoregressive model of suitable order in an iterative way as proposed by Beven *et al*., (2008). Figures 4 c and d, for example, show the use of the residual model fitting visualisations in Workflow 5. The visualisations also serve as an approximate check of the residual model assumptions when analysing posterior simulations.

For the GLUE methods (see Beven and Binley, 1992, 2014; Beven and Freer, 2001; Beven et al., 2008; Beven and Lane, 2022), diagnostics are included for exploration of the acceptable parameter space and which criterion (or criteria) and at which timesteps (or locations) simulations were rejected. There are also method-specific and generic

toolbox functions for visualisation and presentation of simulation results and associated uncertainties (e.g. see

Figure 5 for the application in Workflow 1). Results are both alphanumeric and graphical; alphanumeric results (including those from diagnostic statistics and summary variables where appropriate) can be automatically written to the audit trail log and plots are saved to the project folder.

## 4 Condition Tree Implementation within CURE

An important part of any CURE toolbox application, is the way that users can explore and document modelling choices, assumptions and uncertainties using the condition tree GUI (e.g. Figure 6). The GUI aids in the elicitation of primary modelling uncertainties, their likely sources and how they are to be treated during the analysis. It is also designed to elicit other important choices and assumptions, including those regarding elements of the analysis assumed to be associated with insignificant uncertainties and perhaps treated deterministically; for example, where

only one model structure is considered or where uncertainties are assumed negligible for certain elements or are perhaps subsumed into other uncertain elements. Similar to the incorporation of UE, the condition tree would be completed, ideally, as an integral part of any modelling application and can help in the definition of an appropriate workflow structure.    This is particularly important in considering epistemic sources of uncertainty.  We fully understand that non-probabilistic approaches to uncertainty estimation remain controversial (e.g. Nearing et al.,

2016) but have demonstrated in the past that the assumptions required to use formal statistical methods (e.g. the recent paper of Vrugt et al., 2022) may lead to overconfidence in the resulting inference when epistemic uncertainties are important (Beven and Smith, 2015; Beven, 2016). Because the epistemic uncertainties are the result of lack of knowledge their nature and impacts cannot be defined easily.   That means that effectively there can be no right answer (e.g. Beven et al., 2018a,b; Beven and Lane, 2022) so that the recording of assumptions in

the audit trail for analysis should be a requisite of any analysis to allow later evaluation by others.

The GUI takes the form of a number of simple, sequential dialogue boxes where the user is asked to enter text.  In the initial release of the toolbox there are 5 primary dialogue boxes covering:

1.  Project aims and model(s)/model structures considered
2.  Modelling uncertainties - *overview*: model structure, parameters, inputs, observations for model conditioning

3. Uncertainties - observations for model conditioning - _specific_: associated uncertainties and basis for assessing simulation performance
4. Uncertainties - inputs - _specific_: sampling strategy, distributions, dependencies
5. Uncertainties - parameters - _specific_: choice of parameters, sampling strategy, distributions, dependencies

The information elicited using the dialogue boxes can be automatically written to the project audit trail log during the initial phase of entry; the audit trail log remains editable as the user defines their own workflow and during any subsequent modifications to the analysis contained within a workflow.

## 5 Defining a workflow

An _a priori_ consideration of modelling uncertainties via the condition tree is an _optional_ first step to help choose and structure an appropriate workflow. The decision tree of Figure 1 can also be a guide in this respect. These are complemented by the toolbox _documentation_ and _help text,_ which are available via the workflows and functions. Documentation and help are in the form of targeted comments within the code and function _header text_ is available by typing the _help "function name"_ at the command line (e.g. headers may include a definition of function variables and references for a specific UE method). Each workflow is also linked, where possible, to the relevant chapters of Beven (2009); these are specified in the header text of each workflow script. Clarification of the terminology used in the help and documentation is provided by a glossary of terms included as part of the toolbox.

It is assumed that the user has completed any necessary pre-processing analyses such as forcing-input uncertainty assessment and disinformation screening (e.g. Beven and Smith, 2015) as well as an assessment of uncertainties associated with conditioning observations where used. An exception is the interactive toolbox facility for fitting residual models mentioned earlier when formal statistical likelihoods are to be used.

The example workflows have been chosen to span the UE methods included in the toolbox and, in some cases, provide comparison of different UE methods for similar modelling applications. The structure of the workflows themselves includes the primary steps to be 'populated' as follows:

1. Condition tree GUI: project setup and interactive dialogue boxes
2. Set up inputs and observations

3. Set up parameter ranges, distributions and sampling strategy

4. Define performance measure (if conditioned UE)

5. Simulations (*on-line* or *off-line*; MATLAB™ function or '*external model*')

6. Post-processing: diagnostics, results, propagation and visualisation of uncertainty

Associated with these main steps, example workflows include *automatic* 'text writes' which are appended to the audit trail log for each analysis. These include specific choices which are made when implementing steps 1-5 above: such as the ranges of parameter values used and their distributions, the sampling strategy employed as well as diagnostic and simulation results.

In general, users will not need to modify any toolbox functions; they will only need to build a workflow. However, given the requirement for *on-line* simulation performance to be assessed for MCMC methods, and the many permutations of performance measures, and ways of combining them where multiple criteria are used, users are also required to specify the function that returns an overall measure of individual simulation performance. In addition, where 'external' models are to be used for on-line approaches, additional modifications may be required for modification of input/parameter files, using some form of *wrapper code*.

## 6 An Example Workflow

The CURE Workflows can be applied to a wide range of geoscience applications, including the water science examples set out in Table 1. In particular, it is well suited to the specification of assumptions about epistemic uncertainties, conditioning using uncertain observational data, and rejectionist approaches to model evaluation (see also Beven et al., 2018a,b; 2022a,b). Here we provide some more detail on the application of the PROTECH model within such a multi-variable rejectionist conditioning framework (Workflow 12 in Table 1). The full workflow and outputs are given in the Electronic Supplement to this article.

PROTECH is a lake algal community model that has been applied to predict concentrations for functional classes of algae in Lake Windermere in Cumbria, UK (Page et al., 2017). It is a 1D model with water volumes related to the lake bathymetry and runs with a daily time step. In this case the model is provided in an executable form and was run off-line for randomly sampled parameter sets, so that the workflow takes the simulated output files as

inputs.    The model requires flow, weather and nutrient information as inputs.   A reduced set of six parameters were sampled as in Table 2 (see Page et al., 2017, for a more complete analysis).   Model evaluation is based on limits of acceptability for three variables: chlorophyll, and the concentrations of R-type and CS-type algae.   Figure 7 shows the resulting chlorophyll outputs for the surviving models from the analysis after evaluation against all three sets of limits of acceptability.    The full workflow and resulting audit trail and output figures are presented in the Electronic Supplement.

Table 2.  **Parameters and uniform distribution sampling ranges for the application of the PROTECH model to Lake Windermere (Workflow 12 application).**

| Parameter | Meaning | Min | Max |
|---|---|---|---|
| EPSW | Background light extinction coefficient | 0.15 | 0.35 |
| $P_f$ | Growth rate factor for phosphorus | 0.5 | 2.5 |
| $N_f$ | Growth rate factor for nitrate | 0.5 | 1.5 |
| $Si_f$ | Growth rate factor for silica | 0.5 | 1.5 |
| $K_z$ | Vertical effective eddy diffusion coefficient | 0.05 | 0.4 |
| $WW_f$ | Waste Water Treatment Works adjustment factor for phosphorus | 0.05 | 0.6 |

## 7 Toolbox Evolution

The toolbox structure is such that new methods can be easily added and it will be subject to ongoing development and augmentation with additional workflow examples. It is hoped that the CURE toolbox will contribute to the ongoing development and testing of UE methods and good practice in their application. In particular, the condition tree approach could be further developed via feedback from toolbox users and end-users of the *conditional* uncertainty estimates. The toolbox is freely available for non-commercial research and education from: https://www.lancaster.ac.uk/lec/sites/qnfm/credible.

**Acknowledgements**

This work was supported by the Natural Environment Research Council project: Consortium on Risk in the Environment: Diagnostics, Integration, Benchmarking, Learning and Elicitation (CREDIBLE) grant number NE/J017450/1, and the NERC Q-NFM Project, grant number NE/R004722/1.


## Computer Code Availability

The CURE Matlab Toolbox Version 1.0 is an Open Source Matlab Code hosted at Lancaster University and can be downloaded at https://www.lancaster.ac.uk/lec/sites/qnfm/credible. (contact n.chappell@lancaster.ac.uk).    It was first made available in 2021.


## Author Contributions

Trevor Page and Keith Beven were involved in the conceptualisation of the CURE Toolbox and the development of the example applications. Trevor Page, Paul Smith, Francesca Pianosi and Fanny Sarrazin provided the software development.  Trevor Page and Keith Beven wrote the original draft of the paper. All the other authors were

involved in the development of applications within the NERC CREDIBLE project that was the original motivation for the development of the SAFE and CURE Toolboxes, and in reviewing and editing the paper.  Thorsten Wagener was the principal investigator of CREDIBLE.  The use of the CURE in the NERC Q-NFM project was supported by Nick Chappell who also established the CURE Website.

## Competing Interests

One co-author (JF) is a member of the editorial board of *Hydrology and Earth System Sciences*.

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

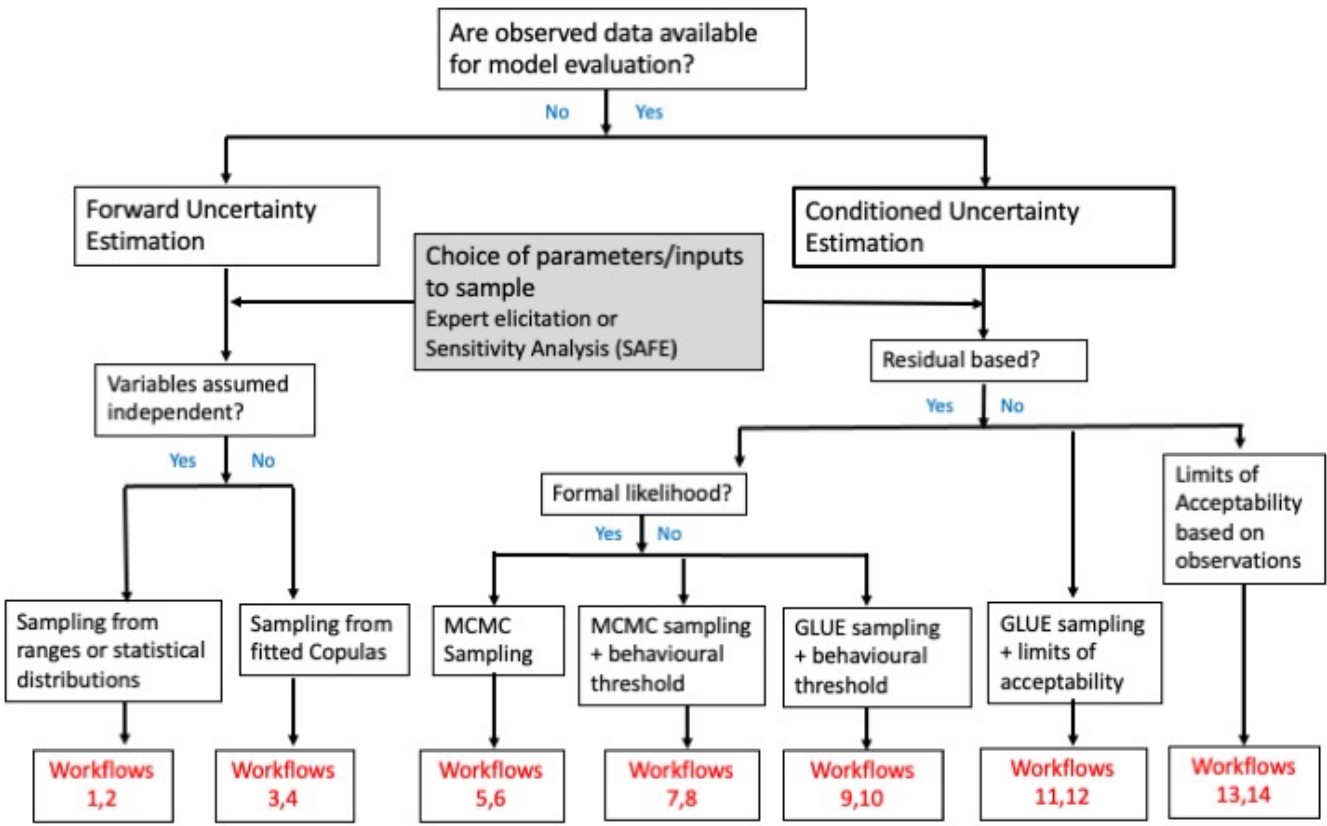

**Figure 1. Decision tree guiding users towards different methodologies and workflows**


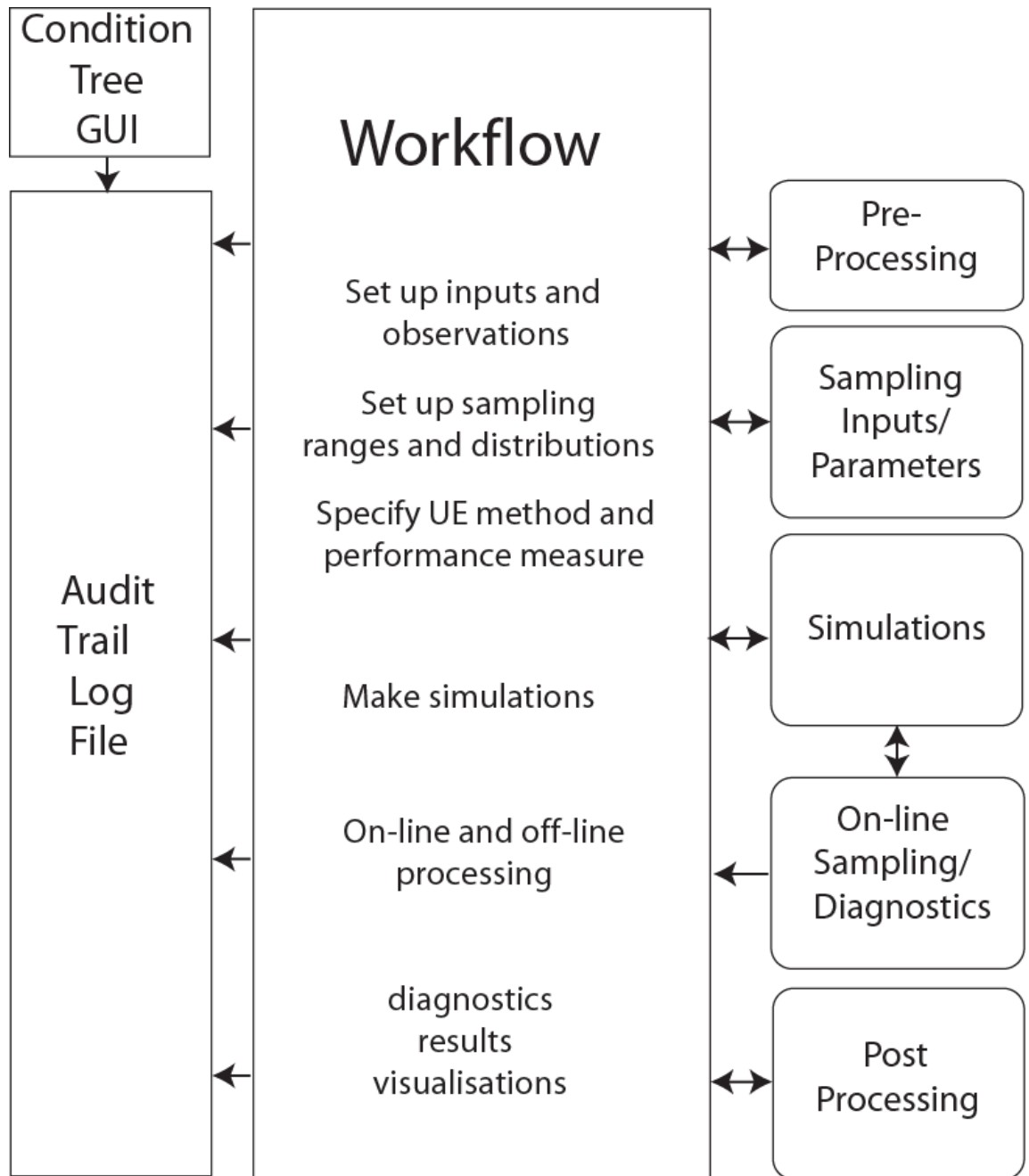

**Figure 2 CREDIBLE Uncertainty Estimation Toolbox: overall structure**

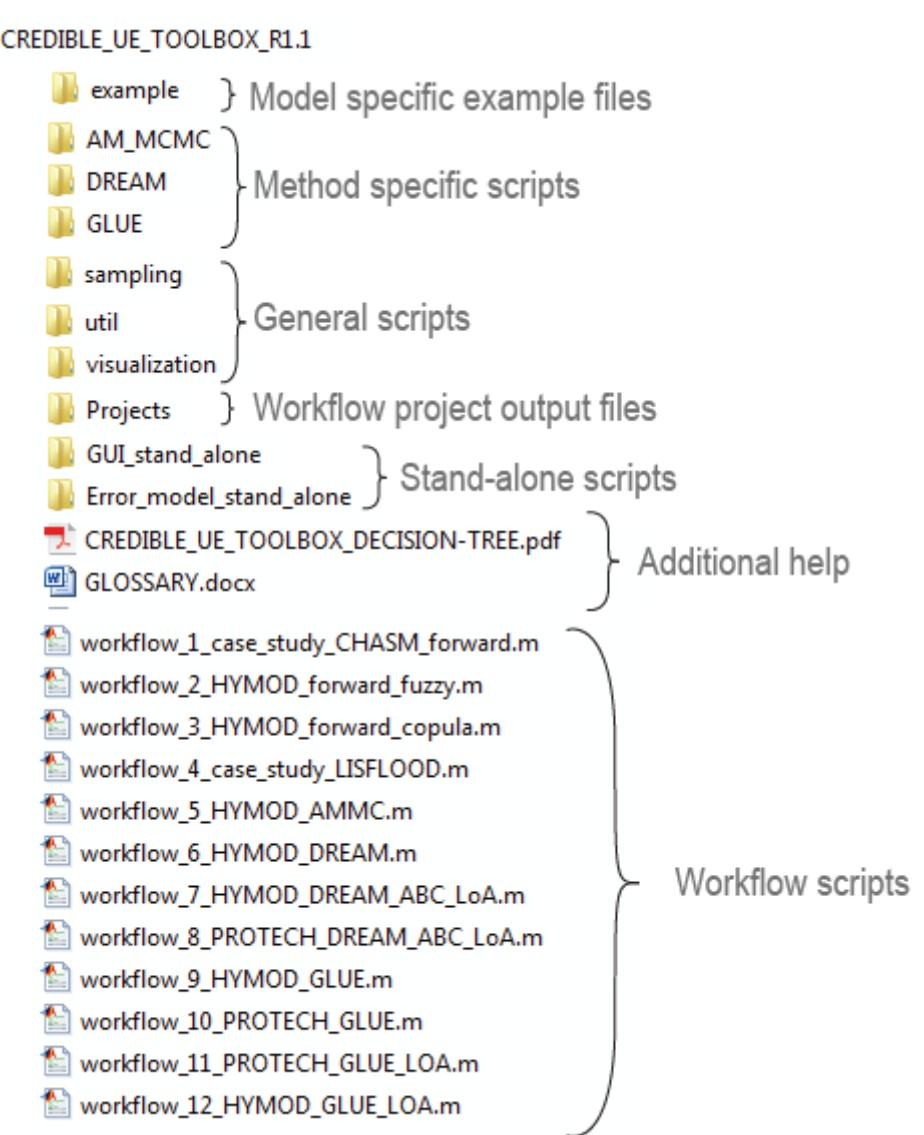

**Figure 3 Outline folder structure of the CURE Toolbox**

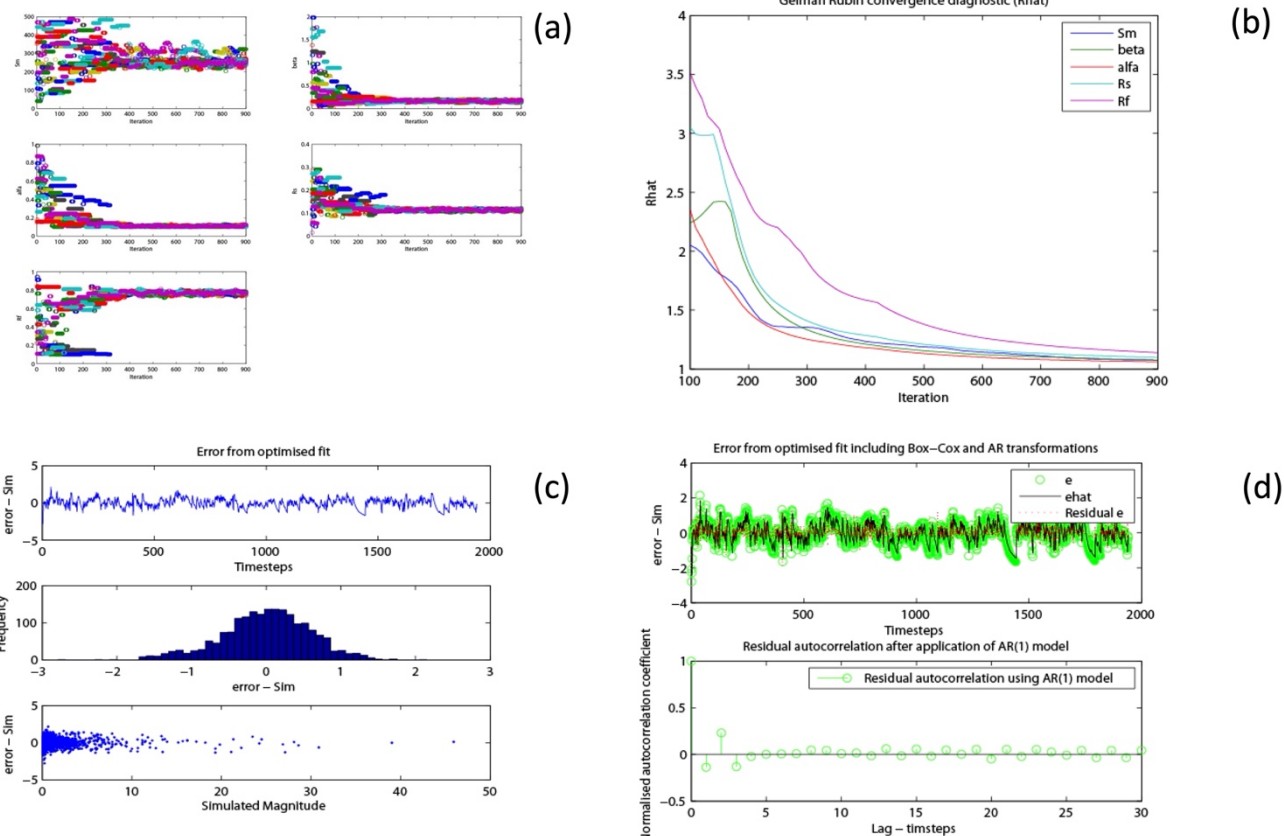

**Figure 4  Visualisation of simulation diagnostics in conditioning of parameters of HYMOD in Workflow 5 using DREAM with a formal likelihood: (a) the evolution of 12 chains using DREAM; (b) evolution of the Gelman Rubin convergence statistic for 5 parameters; (c) & (d) visualisation of  structural parameters during residual model fitting;**

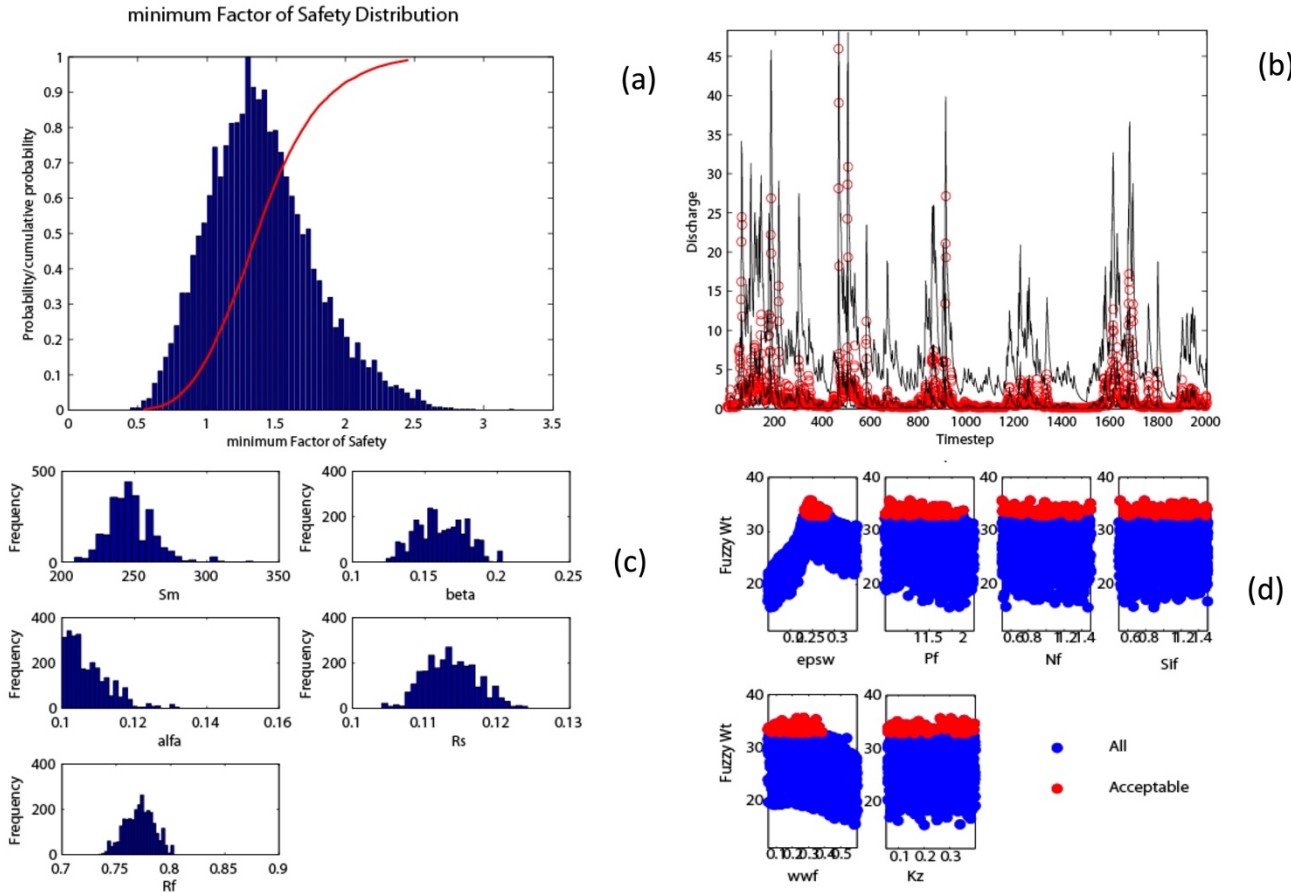

**Figure 5 Visualisation of results: (a) The distribution of simulated minimum factor of safety from a forward UE using the CHASM landslide model in Workflow 1; (b) 5th; 50th and 95th percentiles of simulated discharge (black lines) and observed discharge from an MCMC conditioned UE method using HYMOD; (c) Posterior parameter distributions for the same example as (b) and (d) dotty plots showing both all and acceptable parameter sets from a GLUE analysis using the PROTECH model.**


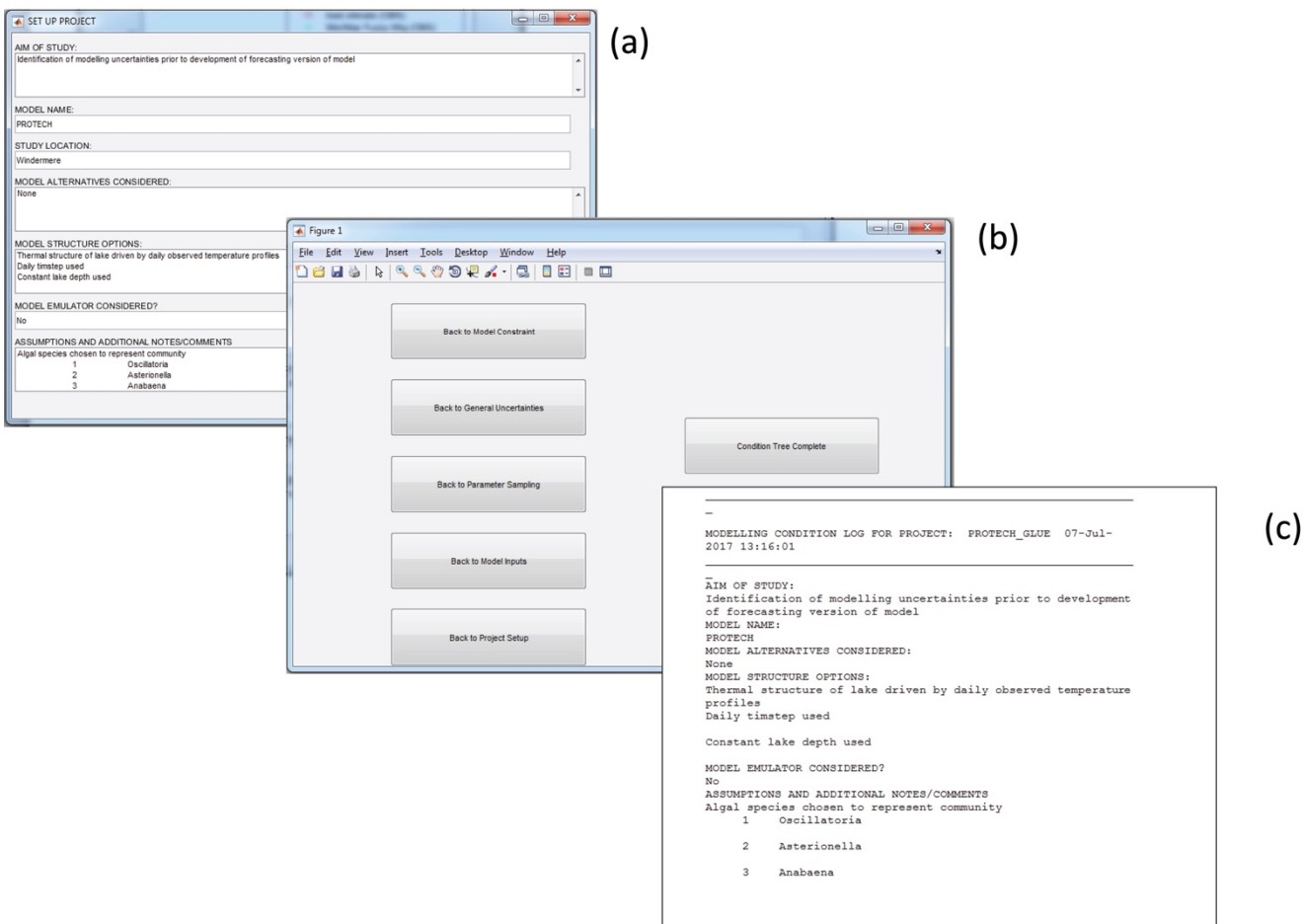

**Figure 6.** Condition Tree example: GUI dialogue box for (a) Project Setup, (b) the Condition Tree navigation pane and (c) part of an example audit trail log.


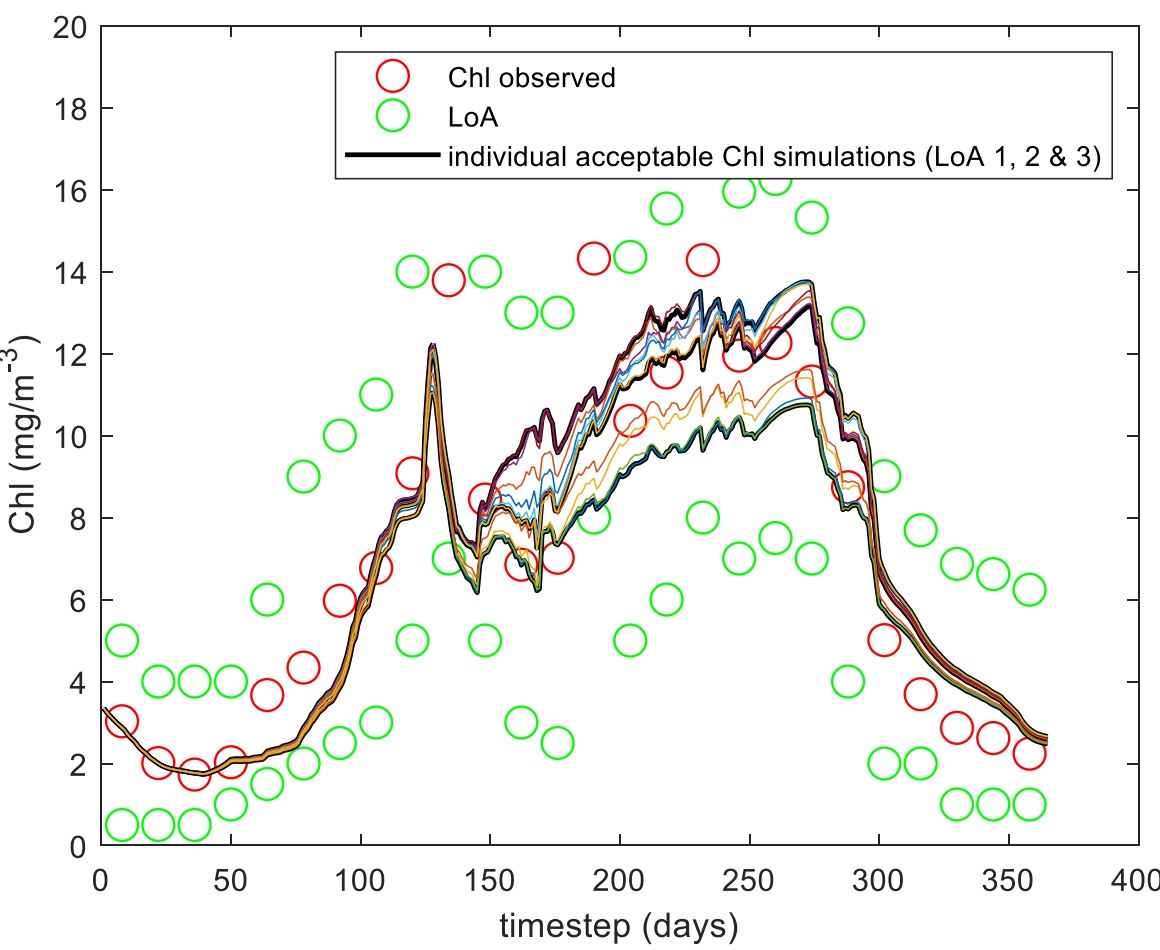


**Figure 7. PROTECH application to Lake Windermere example in Workflow 12: Observed chlorophyll data (red circles), limits of acceptability (green circles) and predictions of models that satisfy all of chlorophyll, R-type and CS-type algae limits.**
