# Peer review of "Technical Note: The CREDIBLE Uncertainty Estimation (CURE) toolbox: facilitating the communication of epistemic uncertainty"

_Hydrology and Earth System Sciences, 2022_

## Referee Comment (RC2)

**Review of the Technical Note "The CREDIBLE Uncertainty Estimation (CURE) toolbox: facilitating the communication of epistemic uncertainty" by Page et al.**

The technical note by Page et al. presents an open-source MATLAB toolbox that combines multiple uncertainty estimation methods. Workflows are provided that exemplify the application of each of these methods. The toolbox also allows the use of a condition tree that enables users to document their choices and assumptions in an attempt to improve transparency and communication of the limitations of the results. While I recognize the value of such a toolbox and appreciate the effort in compiling all these uncertainty estimation methods, I believe a thorough revision of the code and significant improvements to the documentation need to be made. More detailed documentation has the potential to increase the use of the toolbox substantially and may prevent its use in unintended ways (with misleading results/conclusions). I have listed the main thoughts about the paper/code below.

1. **Evaluation against the original implementation**

Given some of the results I saw (for example, the results from Workflow 6), I wonder whether each uncertainty estimation method was tested against their original implementation (i.e., by running the CURE toolbox and the original code using the exact same case study). The authors could choose as case studies for the workflows the examples provided with the original (cited) toolbox (whenever possible), which would allow the user to know the expected results and help interpret the outputs provided by the CURE toolbox.

2. **CURE toolbox vs. original implementation**

I don't believe the codes reflect the implementation described in the references cited in Table 1. For some uncertainty estimation methods, the current implementation is a simplified version of the original code with limited options. This should be explicitly mentioned in the manuscript, and a discussion of potential implications should be incorporated.

3. **Large amount of typos in the codes**

Codes and website content need to be revised carefully. There are multiple typos in the codes that, in some cases, may prevent the user from running a specified configuration (for example, I get an error

when running Workflow 6 by setting "Err_mod_fit = 1" and switching the option "Lifun" from "B_C_AR" to "B_C").

**4. Need to improve the documentation**

I am not sure if the intended audience would be able to use the methods implemented in the toolbox without substantial additional effort.

It is not clear what are the requirements of each uncertainty estimation method (which inputs are needed, what are the user-specified options) or, in other words, which parts of the code the user should modify when creating their own workflow.

A similar comment is valid for the outputs: some output figures do not have appropriate titles/legends; it is not always clear what is the meaning of each figure; and why these figures are being plotted.

It would be helpful to include in the manuscript a table (or something similar) where the required inputs, available options (and what happens when the user selects each option), and outputs of each uncertainty method are at least listed (even better if explanations of the generated outputs are provided).

**5. Treatment of autocorrelation**

I am not sure if the autocorrelation implementation is correct. Any references? I would suggest checking Evin et al. (2013, 2014) and Vrugt et al. (2022).

G. Evin, D. Kavetski, M. Thyer, and G. Kuczera. Pitfalls and improvements in the joint inference of heteroscedasticity and autocorrelation in hydrological model calibration. Water Resources Research, 49(7):4518–4524, 2013.

G. Evin, M. Thyer, D. Kavetski, D. McInerney, and G. Kuczera. Comparison of joint versus postprocessor approaches for hydrological uncertainty estimation accounting for error autocorrelation and heteroscedasticity. Water Resources Research, 50(3):2350–2375, 2014.

Vrugt, J. A., de Oliveira, D. Y., Schoups, G. & Diks, C. G. On the use of distribution-adaptive likelihood functions: Generalized and universal likelihood functions, scoring rules and multi-criteria ranking. Journal of Hydrology 615, 128542. doi:10.1016/j.jhydrol.2022.128542 (2022).

**6. Limitations/advantages of each method and additional guidelines**

Some guidance could be provided for selecting from the different uncertainty estimation methods available. A summary of the limitations/advantages of each method would be helpful, as well as guidelines for the specification of the required parameters (for example, studies like McInerney et al., 2017 could be provided as a reference for helping users to select initial Box-Cox parameter values).

> McInerney, D., Thyer, M., Kavetski, D., Lerat, J., and Kuczera, G. (2017), Improving probabilistic prediction of daily streamflow by identifying Pareto optimal approaches for modeling heteroscedastic residual errors, Water Resour. Res., 53, 2199– 2239, doi:10.1002/2016WR019168.

**7. Going beyond recording assumptions**

The toolbox includes a condition tree that users can use to document choices and assumptions. It would beneficial if the toolbox also included the option of testing (some of) these assumptions. For example, if the uncertainty analysis is conducted using a formal likelihood/Bayesian inference, common checks would be to evaluate the residual assumptions and the quality of the uncertainty estimates (see Thyer et al., 2009 and many others).

> M. Thyer, B. Renard, D. Kavetski, G. Kuczera, S. W. Franks, and S. Srikanthan. Critical evaluation of parameter consistency and predictive uncertainty in hydrological modeling: A case study using Bayesian total error analysis. Water Resources Research, 45(12), 2009.

While this might be beyond the scope of the paper, at least some discussion on how the assumptions can be tested could be included, especially for "informal" metrics (given that in some cases the assumptions are not known, how can we evaluate the reliability of the results?).

**Minor comments**

1. What is the difference between the GLUE-Limits of Acceptability (LoA) method and the use of LoA within the DREAM toolbox (Vrugt & Beven, 2018)?

> J. A. Vrugt and K. J. Beven. Embracing equifinality with efficiency: Limits of acceptability sampling using the DREAM(LOA) algorithm. Journal of Hydrology, 559:954–971, 2018

2. McInerney et al. (2018) could be added to the introduction.

D. McInerney, M. Thyer, D. Kavetski, B. Bennett, J. Lerat, M. Gibbs, and G. Kuczera. A simplified approach to produce probabilistic hydrological model predictions. Environmental Modelling Software, 109:306–314, 2018.

3. Residuals and errors are used interchangeably, but they do not have the same meaning. From Vrugt & de Oliveira (2022): "The word error implies a difference between an observed value and its true value. As our measurements of system behavior are imperfect, the residuals are estimates of the errors under the assumed model. Hence, we should use the word residual instead."

Vrugt, J. A. & de Oliveira, D. Y. Confidence intervals of the Kling-Gupta efficiency. Journal of Hydrology 612, 127968. doi:10.1016/j.jhydrol.2022.127968 (2022).

4. Glossary: some definitions are somewhat informal and not necessarily accurate. Even though the toolbox was designed for individuals "who are not necessarily experts in uncertainty estimation", it is important to define the terms accurately.

5. Code, Workflow 1: "This Workflow is a CASE STUDY demonstrating forward uncertainty analysis for the CHASM landslide model (see CHASM_IO_files_2014.pdf in CHASM folder)": I couldn't find the "CHASM_IO_files_2014.pdf" file

6. What is the difference between a "case study" (workflow) and "application example"/"example" (other workflows)?

7. Website, Learn about CURE and Uncertainty Estimation: Update link and reference to Page et al. (2021)

8. Website, Case Studies: Why only 4, 5, and 11 are presented here?

---

## Author Comment (AC1)

**Referee 1: Tobi Krueger**

This is a short technical note on the first release of an uncertainty estimation toolbox for Matlab. The toolbox itself has very useful features, most notably from my perspective: advanced multivariate distributional assumptions through copulas; an audit trail log of assumptions made in the process; and 12 workflow scripts that will help the analyst apply the toolbox.

In my opinion it would increase the value of the paper (as an addition to the documentation of the toolbox) if the authors could give a little more detail on the nature of the uncertainty problem in each workflow. Now, if the paper is the first port of call for those wanting to use the toolbox, they have to do a lot of additional background reading before they can decide which method to use or which workflow to follow. If the authors could give a little more detail on the nature of uncertainties considered in each workflow and why this led them to use a certain method and not another. This would then give the reader a sense of which method is appropriate for what.

The organisation of the paper has been revised to give more prominence to the discussion of methods and choice of workflow (former Figure 6) earlier in the paper.

The description of the toolbox is otherwise clear, with the exception of the handling of input data uncertainty in the "conditioned" case. See also my comment to Fig6 below: Is input data uncertainty handled the same way as parameter priors? There should be added complexity due to the timeseries nature of inputs. What are the choices available to the analyst here? I'm thinking of rainfall multipliers and various other approaches that have been suggested. And are these prior choices of input data uncertainty updated conditional on the observations of model output? I think here we need more information in the paper.

The text and Table 1 have now been revised to make the choices clearer.

Specific comments:

There is a font size change from L75 onwards. Is this intentional?

No, this has now been changed

When reviewing the other toolboxes from L75 onwards, could the authors flag those toolboxes that are still maintained? With some I had the feeling they might not be (certainly DUE) and then they are of limited value in my opinion.

That is a good point.   We have added a comment that this is the case, though it is not always possible to tell from the sites as to whether they are still maintained or not.

On the CURE website there is still what seems like an older version of the paper submitted to EMS. This should be updated.

This will be updated when the revised paper is submitted

Fig3-4: The font sizes and tick labels (overlap) could be improved in some instances.

This has been corrected

Fig3: I find the evolution plot of the Rhat statistic not so useful since we can't see the area around 1 very well at that scale where we want Rhat to end up. I encourage the authors to consider removing this plot or finding a way to scale it better (I guess the final Rhat value is reported in any case, which would be important).

These was a result of the way in which the figure was displayed. It is now improved.

Fig6: I'm unclear about the two branches coming from "Conditioned Uncertainty Analysis". Is the analyst meant to go down both branches? If so, this could be made clear. This way the analyst seems to end up with prior parameter distributions at the end of the left branch that then feed into the sampling strategies emerging from the right branch. But what about the multivariate cases – what do they lead to? The same could be said about the middle arrow on the very left (forward uncertainty analysis). More importantly, what about input data uncertainty? Is this handled the same way as parameter priors? There should be added complexity due to the timeseries nature of inputs. What are the choices available to the analyst here? I'm thinking of rainfall multipliers and various other approaches that have been suggested. And are these prior choices of input data uncertainty updated conditional on the observations of model output?

This is certainly ambiguous and has now been clarified in the text and in a revised Figure 6 which now makes those two branches clearer.

Comments for general discussion:

I'm here noting a few general discussion points that I invite that the authors to engage with, though they are not of central importance to the paper.

First, I want to encourage the authors to eventually publish their toolbox for an open-source software environment like R as well. Or at least comment on the compatibility of the toolbox with clones like GNU Octave.

We would certainly like to do so (as with the earlier highly successful SAFE Toolbox, Pianosi et al EMS 2016), given time. If it is possible to add other versions then they will be made available as open source on the CURE site.

Pianosi, F., J. Rougier, J. Freer, J. Hall, D. B. Stephenson, K. J. Beven, and T. Wagener, 2016, Sensitivity Analysis of environmental models: a systematic review with practical workflows, *Environmental Modelling and Software,* 79: 214-232

Second, I'm increasingly wondering whether the distinction between formal and informal uncertainty methods is really productive (I say this having used this distinction myself). This terminology once served to circumvent accusations of incoherence of methods like GLUE just by introducing a different label (informal), but now distracts from engaging with what really matters: the assumptions made about various sources of uncertainty when using certain methods (a problem that the authors summarise well by the way and tackle through their audit trail). With the formal/informal terminology one is led to believe one has a choice between formal and informal methods, while in reality in both cases one has a much more difficult choice of how exactly to model uncertainties and how to aggregate individual (e.g. timestep-based) performance measures into an overall metric (e.g. multiplicative or additive) and what understanding of uncertainties and model performance this entails (often implicitly). Such understanding goes down to foundational axioms as discussed by Nearing et al. (2016), which the authors cite. Any

formal/informal distinction would also be increasingly blurred by methods such as ABC –
here I'm missing reference to work by Lucy Marshall and co-workers discussing the
similarities between ABC and GLUE, by the way. Maybe the authors can re-evaluate
their use of these terms and engage in a broader discussion.

This is an important point, but, we would suggest, not as clear-cut as Nearing et al
suggest.   Yes, if you are happy to fit into the axioms of probability theory then there is a
fundamental foundation to all that follows.   But that assumes that the probabilities are
complete and non-conflicting (the excluded middle).  One way round this is to allow
alternative probabilistic representations in parallel (e.g. Rougier and Beven, 2013) but
there are alternative sets of axioms that could be considered, such as those underlying
fuzzy set theory (e.g. the Halpen 2003 book).  The point we wish to make is that there
are alternative assumptions (within broadly being Bayesian in conditioning the outputs
by information) but that the assumptions made must be made explicit since, for the case
of epistemic uncertainties, there can be no right answer (Beven et al., NHESS
2018a,b).   We would suggest that Bayes' use of subjective odds in his original paper, is
compatible with this approach without the need for formal likelihood reasoning.

Rougier, J and Beven, K J, 2013, Model limitations: the sources and implications of epistemic uncertainty, in
Rougier J, Sparks, S and Hill, L (Eds), *Risk and uncertainty assessment for natural hazards*, Cambridge
University Press: Cambridge, UK, 40-63

Beven, K. J., Almeida, S., Aspinall, W. P., Bates, P. D., Blazkova, S., Borgomeo, E., Freer, J., Goda, K., Hall, J.
W., Phillips, J. C., Simpson, M., Smith, P. J., Stephenson, D. B., Wagener, T., Watson, M., and Wilkins, K. L.,
2018, Epistemic uncertainties and natural hazard risk assessment.  1.  A review of different natural hazard areas,
*Natural Hazards and Earth System Science,* 18(10): 2741-2768. https://doi.org/10.5194/nhess-18-2741-2018

Beven, K J, Aspinall, W P, Bates, P D, Borgomeo, E, Goda, K, Hall, J W, Page, T, Phillips, J C, Simpson, M,
Smith, P J, Wagener, T and Watson, M, 2018, Epistemic uncertainties and natural hazard risk assessment – Part
2: What should constitute good practice?, *Natural Hazards and Earth System Science*, , 18(10): 2769-2783,
https://doi.org/10.5194/nhess-18-2769-2018

Third, I find the comment about rigorous statistical treatment of aleatory but not
epistemic uncertainty in L59-61 misleading. This seems to be relating more to
differences between frequentist and Bayesian statistical methods than anything else.
The Bayesian framework deals expressly with epistemic uncertainties. The question is
just whether or not the assumptions one makes are justified – but this is the case with
any uncertainty method (which the authors emphasise well in this paper). I encourage
the authors to remove this reference to epistemic versus aleatory uncertainty and focus
on the importance of choices (which they already do) – in any method.

Again we are emphasising that importance of making assumptions explicit.   But there is
a difference.  Under aleatory assumptions you can develop a formal likelihood function
for use within Bayes equation.   But we know that for many cases of practical interest
where epistemic uncertainties are important this gives misleading results by stretching
the likelihood surface (in some cases of time series enormously).   Although those
favouring the probabilistic approach might suggest that it is then the definition of the
likelihood that needs improving (e.g. the recent paper of Vrugt et al., JH2022 suggesting
a "universal" likelihood that still ignores sources of epistemic uncertainty except in so far
as they affect the statistics of the residuals).   But, we would suggest that it might be
possible to be more thoughtful than that, for example in trying to use knowledge about
the observations to define limits of acceptability, eg. The recent paper by Beven et al.
(HP2022).

Vrugt, J.A., de Oliveira, D.Y., Schoups, G. and Diks, C.G., 2022. On the use of distribution-adaptive likelihood functions: Generalized and universal likelihood functions, scoring rules and multi-criteria ranking. *Journal of Hydrology*, *615*, p.128542.

Beven, K. J., Lane, S., Page, T., Hankin, B, Kretzschmar, A., Smith, P. J., Chappell, N., 2022, On (in)validating environmental models. 2. Implementation of the Turing-like Test to modelling hydrological processes, *Hydrological Processes*, 36(10), e14703, **https://doi.org/10.1002/hyp.14703**.

---

## Author Comment (AC2)

**Response to Comments on CURE Paper**

**Referee 2.**

**Review of the Technical Note "The CREDIBLE Uncertainty Estimation (CURE) toolbox: facilitating the communication of epistemic uncertainty" by Page et al.**
The technical note by Page et al. presents an open-source MATLAB toolbox that combines multiple uncertainty estimation methods. Workflows are provided that exemplify the application of each of these methods. The toolbox also allows the use of a condition tree that enables users to document their choices and assumptions in an attempt to improve transparency and communication of the limitations of the results. While I recognize the value of such a toolbox and appreciate the effort in compiling all these uncertainty estimation methods, I believe a thorough revision of the code and significant improvements to the documentation need to be made. More detailed documentation has the potential to increase the use of the toolbox substantially and may prevent its use in unintended ways (with misleading results/conclusions). I have listed the main thoughts about the paper/code below.

**1. Evaluation against the original implementation**
Given some of the results I saw (for example, the results from Workflow 6), I wonder whether each uncertainty estimation method was tested against their original implementation (i.e., by running the CURE toolbox and the original code using the exact same case study). The authors could choose as case studies for the workflows the examples provided with the original (cited) toolbox (whenever possible), which would allow the user to know the expected results and help interpret the outputs provided by the CURE toolbox.

Our example case studies are simplifications of the original studies in some cases and hypothetical examples in others. The cited papers for the original studies still allow users to interpret the results. We have carried out extensive testing of the workflows to ensure they give sensible results. We agree it might have been a good idea to present a comparison of common results, but we naturally wished to use results from our own projects to illustrate the different methods.

**2. CURE toolbox vs. original implementation**
I don't believe the codes reflect the implementation described in the references cited in Table 1. For some uncertainty estimation methods, the current implementation is a simplified version of the original code with limited options. This should be explicitly mentioned in the manuscript, and a discussion of potential implications should be incorporated.

The referee is correct and this will be made clear in the revised manuscript. We did not intentionally simplify any of the code. The Adaptive Metropolis MCMC examples were implemented by the authors and were used for several years for teaching purposes. With respect to DREAM, the code was implemented directly from the p-code provided in the 2015 DREAM Manual found on the DREAM website. At the time (the CURE Toolbox has been a long time in development) this was the version readily available and is illustrative for the type of

Vrugt, J, 2015, *Markov chain Monte Carlo Simulation Using the DREAM Software Package: Theory, Concepts, and MATLAB Implementation.*

**3. Large amount of typos in the codes**

Codes and website content need to be revised carefully. There are multiple typos in the codes that, in some cases, may prevent the user from running a specified configuration (for example, I get an error
when running Workflow 6 by setting "Err_mod_fit = 1" and switching the option "Lifun" from "B_C_AR" to "B_C").

The referee is correct.   A typo caused a flag not to be recognised.   This has been corrected and the workflow codes are being checked for additional typos in both code and comments.

**4. Need to improve the documentation**

I am not sure if the intended audience would be able to use the methods implemented in the toolbox without substantial additional effort.
It is not clear what are the requirements of each uncertainty estimation method (which inputs are needed, what are the user-specified options) or, in other words, which parts of the code the user should modify when creating their own workflow.
A similar comment is valid for the outputs: some output figures do not have appropriate titles/legends; it is not always clear what is the meaning of each figure; and why these figures are being plotted.
It would be helpful to include in the manuscript a table (or something similar) where the required inputs, available options (and what happens when the user selects each option), and outputs of each uncertainty method are at least listed (even better if explanations of the generated outputs are provided).

We have taken this on board.   More comments have been added to clarify the inputs and options available and the significance of the outputs.   More information has also been added in restructuring the paper to say more about choice of methods.    It has been made clear that the examples are illustrative, and that this paper is presented as a Technical Note and cannot include all the detail.

**5. Treatment of autocorrelation**

I am not sure if the autocorrelation implementation is correct. Any references? I would suggest checking Evin et al. (2013, 2014) and Vrugt et al. (2022).

G. Evin, D. Kavetski, M. Thyer, and G. Kuczera. Pitfalls and improvements in the joint inference of heteroscedasticity and autocorrelation in hydrological model calibration. Water Resources Research, 49(7):4518–4524, 2013.
G. Evin, M. Thyer, D. Kavetski, D. McInerney, and G. Kuczera. Comparison of joint versus postprocessor approaches for hydrological uncertainty estimation accounting for error autocorrelation and heteroscedasticity. Water Resources Research, 50(3):2350–2375, 2014.
Vrugt, J. A., de Oliveira, D. Y., Schoups, G. & Diks, C. G. On the use of distribution-adaptive likelihood functions: Generalized and universal likelihood functions, scoring rules and multi-criteria ranking. Journal of Hydrology 615, 128542. doi:10.1016/j.jhydrol.2022.128542 (2022).

This has been checked and the implementation is correct.

**6. Limitations/advantages of each method and additional guidelines**

Some guidance could be provided for selecting from the different uncertainty estimation methods available. A summary of the limitations/advantages of each method would be helpful, as well as guidelines for the specification of the required parameters (for example, studies like McInerney et al., 2017 could be provided as a reference for helping users to select initial Box-Cox parameter values).

McInerney, D., Thyer, M., Kavetski, D., Lerat, J., and Kuczera, G. (2017), Improving probabilistic prediction of daily streamflow by identifying Pareto optimal approaches for modeling heteroscedastic residual errors, Water Resour. Res., 53, 2199– 2239, doi:10.1002/2016WR019168.

The discussion of choice of workflow in the paper is being revised and brought forward.   In additional, as noted above, more comments have been added to the workflows to clarify input options and outputs.

**7. Going beyond recording assumptions**

The toolbox includes a condition tree that users can use to document choices and assumptions. It would beneficial if the toolbox also included the option of testing (some of) these assumptions. For example, if the uncertainty analysis is conducted using a formal likelihood/Bayesian inference, common checks would be to evaluate the residual assumptions and the quality of the uncertainty estimates (see Thyer et al., 2009 and many others).

M. Thyer, B. Renard, D. Kavetski, G. Kuczera, S. W. Franks, and S. Srikanthan. Critical evaluation of parameter consistency and predictive uncertainty in hydrological modeling: A case study using Bayesian total error analysis. Water Resources Research, 45(12), 2009.

While this might be beyond the scope of the paper, at least some discussion on how the assumptions can be tested could be included, especially for "informal" metrics (given that in some cases the assumptions are not known, how can we evaluate the reliability of the results?).

A section will be added on testing of assumptions.   Clearly this can be done more formally for some workflows than others.   In dealing with epistemic uncertainty there can be no right answer so the assumptions will be necessarily subjective (we have added references to recent papers discussing this).   However, we have added non-parametric checks on normality with outputs for the workflows using the Box-Cox transformation.

**Minor comments**

1. What is the difference between the GLUE-Limits of Acceptability (LoA) method and the use of LoA within the DREAM toolbox (Vrugt & Beven, 2018)?

J. A. Vrugt and K. J. Beven. Embracing equifinality with efficiency: Limits of acceptability sampling using the DREAM(LOA) algorithm. Journal of Hydrology, 559:954–971, 2018

This is now discussed

2. McInerney et al. (2018) could be added to the introduction.

D. McInerney, M. Thyer, D. Kavetski, B. Bennett, J. Lerat, M. Gibbs, and G. Kuczera. A simplified approach to produce probabilistic hydrological model predictions. Environmental Modelling Software, 109:306–314, 2018.

This has been done.

3.      Residuals and errors are used interchangeably, but they do not have the same meaning. From Vrugt & de Oliveira (2022): "The word error implies a difference between an observed value and its true value. As our measurements of system behavior are imperfect, the residuals are estimates of the errors under the assumed model. Hence, we should use the word residual instead."
4.

This is correct.   The revision is much more careful in using these terms

> Vrugt, J. A. & de Oliveira, D. Y. Confidence intervals of the Kling-Gupta efficiency. Journal of Hydrology 612, 127968. doi:10.1016/j.jhydrol.2022.127968 (2022).

4.      Glossary: some definitions are somewhat informal and not necessarily accurate. Even though the toolbox was designed for individuals "who are not necessarily experts in uncertainty estimation", it is important to define the terms accurately.

The Glossary definitions have been reviewed and revised

5.      Code, Workflow 1: "This Workflow is a CASE STUDY demonstrating forward uncertainty analysis for the CHASM landslide model (see CHASM_IO_files_2014.pdf in CHASM folder)": I couldn't find the "CHASM_IO_files_2014.pdf" file

This has been corrected

6.      What is the difference between a "case study" (workflow) and "application example"/"example" (other workflows)?

Usage of case study / workflow / application has been reviewed throughout

7.      Website, Learn about CURE and Uncertainty Estimation: Update link and reference to Page et al. (2021)

This has now been modified

8.      Website, Case Studies: Why only 4, 5, and 11 are presented here?

These were intended to serve as examples.   Further Case Studies will be added to the website, in particular we now have papers published since this Technical Note was prepared that relate to Workflows 13 and 14..

---

## Author Response (AR1)

**Response to Comments on CURE Paper**

**Editor's Comments**

I have read the paper with interest and I believe that the tool here presented represents a useful reference for the hydrological community, as it integrates the work of some of the authors who most actively have developed sensitivity and uncertainty estimation methods in hydrology.
I also would like to thank the two reviewers that took the time to provide a detailed review, also in some cases going into the organization of the code and checking that everything runs smoothly. I checked a few of the Matlab workflows provided, which were running smoothly.
As a minor comment, I would like to add that the authors should include a version number, which the paper is referring to. This should also correspond to the toolbox website. This is a requirement for GMD (the Copernicus journal for software), but it makes sense to add it also here.
I invite the authors to provide a revised version together with a response that explains how the reviewers' comments have been addressed.

Thanks for the positive comments.   We have indeed added a Version Number both in the paper and on the Website.

We have also reorganised the paper and clarified many points in response to the comments from the 2 referees.    The choice of method is now given much greater discussion towards the start of the paper (now Section 2) and Figure 6 (the new Figure 1) has been modified to be clearer.

Our final responses are detailed below.

**Referee 1: Tobi Krueger**

This is a short technical note on the first release of an uncertainty estimation toolbox for Matlab. The toolbox itself has very useful features, most notably from my perspective: advanced multivariate distributional assumptions through copulas; an audit trail log of assumptions made in the process; and 12 workflow scripts that will help the analyst apply the toolbox.

In my opinion it would increase the value of the paper (as an addition to the documentation of the toolbox) if the authors could give a little more detail on the nature of the uncertainty problem in each workflow. Now, if the paper is the first port of call for those wanting to use the toolbox, they have to do a lot of additional background reading before they can decide which method to use or which workflow to follow. If the authors could give a little more detail on the nature of uncertainties considered in each workflow and why this led them to use a certain method and not

another. This would then give the reader a sense of which method is appropriate for what.

More on sources of uncertainty and choice of methods has been added.   The material has also been reorganised to discuss the choice of method prior to presenting the detail.   We think this makes the options clearer.

The description of the toolbox is otherwise clear, with the exception of the handling of input data uncertainty in the "conditioned" case. See also my comment to Fig6 below: Is input data uncertainty handled the same way as parameter priors? There should be added complexity due to the timeseries nature of inputs. What are the choices available to the analyst here? I'm thinking of rainfall multipliers and various other approaches that have been suggested. And are these prior choices of input data uncertainty updated conditional on the observations of model output? I think here we need more information in the paper.

The discussion of input uncertainty has been made clearer.   The referee is correct that in the workflows presented this has been parameterised.

Specific comments:

There is a font size change from L75 onwards. Is this intentional?

No, this has now been changed

When reviewing the other toolboxes from L75 onwards, could the authors flag those toolboxes that are still maintained? With some I had the feeling they might not be (certainly DUE) and then they are of limited value in my opinion.

That is a good point.   We have added a comment that this is the case, though it is not always possible to tell from the sites as to whether they are still maintained or not.

On the CURE website there is still what seems like an older version of the paper submitted to EMS. This should be updated.

This has now been updated

Fig3-4: The font sizes and tick labels (overlap) could be improved in some instances.

Fig3: I find the evolution plot of the Rhat statistic not so useful since we can't see the area around 1 very well at that scale where we want Rhat to end up. I encourage the authors to consider removing this plot or finding a way to scale it better (I guess the final Rhat value is reported in any case, which would be important).

These figures have been improved.

Fig6: I'm unclear about the two branches coming from "Conditioned Uncertainty Analysis". Is the analyst meant to go down both branches? If so, this could be made clear. This way the analyst seems to end up with prior parameter distributions at the end of the left branch that then feed into the sampling strategies emerging from the right branch. But what about the multivariate cases – what do they lead to? The same could be said about the middle arrow on the very left (forward uncertainty analysis). More importantly, what about input data uncertainty? Is this handled the same way as parameter priors? There should be added complexity due to the timeseries nature of inputs. What are the choices available to the analyst here? I'm thinking of rainfall multipliers and various other approaches that have been suggested. And are these prior choices of input data uncertainty updated conditional on the observations of model output?

This has now been clarified in the reorganised text and the new figure 1

Comments for general discussion:

I'm here noting a few general discussion points that I invite that the authors to engage with, though they are not of central importance to the paper.

First, I want to encourage the authors to eventually publish their toolbox for an open-source software environment like R as well. Or at least comment on the compatibility of the toolbox with clones like GNU Octave.

We would certainly like to do so, given time.   If it is possible to add other versions then they will be made available as open source on the CURE site (this has previously been done with the related SAFE toolbox).

Second, I'm increasingly wondering whether the distinction between formal and informal uncertainty methods is really productive (I say this having used this distinction myself). This terminology once served to circumvent accusations of incoherence of methods like GLUE just by introducing a different label (informal), but now distracts from engaging with what really matters: the assumptions made about various sources of uncertainty when using certain methods (a problem that the authors summarise well by the way and tackle through their audit trail). With the formal/informal terminology one is led to believe one has a choice between formal and informal methods, while in reality in both cases one has a much more difficult choice of how exactly to model uncertainties and how to aggregate individual (e.g. timestep-based) performance measures into an overall metric (e.g. multiplicative or additive) and what understanding of uncertainties and model performance this entails (often implicitly). Such understanding goes down to foundational axioms as discussed by Nearing et al. (2016), which the authors cite. Any formal/informal distinction would also be increasingly blurred by methods such as ABC – here I'm

missing reference to work by Lucy Marshall and co-workers discussing the similarities between ABC and GLUE, by the way. Maybe the authors can re-evaluate their use of these terms and engage in a broader discussion.

This is an important point, but, we would suggest, not as clear-cut as Nearing et al suggest.   Yes, if you are happy to fit into the axioms of probability theory then there is a fundamental foundation to all that follows.   But that assumes that the probabilities are complete and non-conflicting (the excluded middle).  One way round this is to allow alternative probabilistic representations in parallel (e.g. Rougier and Beven, 2013) but there are alternative sets of axioms that could be considered, such as those underlying fuzzy set theory (e.g. the Halpen 2003 book). The point we wish to make is that there are alternative assumptions (within broadly being Bayesian in conditioning the outputs by information) but that the assumptions made must be made explicit since, for the case of epistemic uncertainties, there can be no right answer.   We would suggest that Bayes' use of subjective odds in his original paper, is compatible with this approach without the need for formal likelihood reasoning.

Third, I find the comment about rigorous statistical treatment of aleatory but not epistemic uncertainty in L59-61 misleading. This seems to be relating more to differences between frequentist and Bayesian statistical methods than anything else. The Bayesian framework deals expressly with epistemic uncertainties. The question is just whether or not the assumptions one makes are justified – but this is the case with any uncertainty method (which the authors emphasise well in this paper). I encourage the authors to remove this reference to epistemic versus aleatory uncertainty and focus on the importance of choices (which they already do) – in any method.

Again we are emphasising that importance of making assumptions explicit.   But there is a difference.  Under aleatory assumptions you can develop a formal likelihood function for use within Bayes equation.   But we know that for many cases of practical interest where epistemic uncertainties are important this gives misleading results by stretching the likelihood surface (in some cases of time series enormously).   Although those favouring the probabilistic approach might suggest that it is then the definition of the likelihood that needs improving (e.g. the recent paper of Vrugt et al., JH2022 suggesting a "universal" likelihood that still ignores sources of epistemic uncertainty except in so far as they affect the statistics of the residuals).   But, we would suggest that it might be possible to be more thoughtful than that, for example in trying to use knowledge about the observations to define limits of acceptability, eg. The recent paper by Beven et al. HP2022).

**Referee 2.**

**Review of the Technical Note "The CREDIBLE Uncertainty Estimation (CURE) toolbox: facilitating the communication of epistemic uncertainty" by Page et al.**

The technical note by Page et al. presents an open-source MATLAB toolbox that combines multiple uncertainty estimation methods. Workflows are provided that exemplify the application of each of these methods. The toolbox also allows the use of a condition tree that enables users to document their choices and assumptions in an attempt to improve transparency and communication of the limitations of the results. While I recognize the value of such a toolbox and appreciate the effort in compiling all these uncertainty estimation methods, I believe a thorough revision of the code and significant improvements to the documentation need to be made. More detailed documentation has the potential to increase the use of the toolbox substantially and may prevent its use in unintended ways (with misleading results/conclusions). I have listed the main thoughts about the paper/code below.

**1. Evaluation against the original implementation**

Given some of the results I saw (for example, the results from Workflow 6), I wonder whether each uncertainty estimation method was tested against their original implementation (i.e., by running the CURE toolbox and the original code using the exact same case study). The authors could choose as case studies for the workflows the examples provided with the original (cited) toolbox (whenever possible), which would allow the user to know the expected results and help interpret the outputs provided by the CURE toolbox.

At the time the workflows were implemented many tests were done to check whether expected results were obtained.  These results have not been saved, but we feel secure in the methods as implemented.  In some cases, what is presented in the toolbox is a simplification of earlier applications, to aid clarity.  The workflows are only intended to be illustrative of the methods, it made clear that it is up to the user to modify a workflow for their own needs. The comments in each workflow are intended to aid this.

**2. CURE toolbox vs. original implementation**

I don't believe the codes reflect the implementation described in the references cited in Table 1. For some uncertainty estimation methods, the current implementation is a simplified version of the original code with limited options. This should be explicitly mentioned in the manuscript, and a discussion of potential implications should be incorporated.

This is now mentioned.

**3. Large amount of typos in the codes**

Codes and website content need to be revised carefully. There are multiple typos in the codes that, in some cases, may prevent the user from running a specified configuration (for example, I get an error

when running Workflow 6 by setting "Err_mod_fit = 1" and switching the option "Lifun" from "B_C_AR" to "B_C").

The referee is correct.  This has now been corrected and the comments in all the workflows have been reviewed.

**4. Need to improve the documentation**

I am not sure if the intended audience would be able to use the methods implemented in the toolbox without substantial additional effort.

We think that this underestimates the potential users. The equivalent and related SAFE Toolbox has been extensively downloaded and widely used and cited but also requires effort to be adapted to new applications. The extensive comments in each workflow should help in this respect. The GLUE workflow methods in particular are relatively easy to understand and modify (and have been discussed extensively in the literature)

It is not clear what are the requirements of each uncertainty estimation method (which inputs are needed, what are the user-specified options) or, in other words, which parts of the code the user should modify when creating their own workflow.

Again, the comments in each workflow help make this clear

A similar comment is valid for the outputs: some output figures do not have appropriate titles/legends; it is not always clear what is the meaning of each figure; and why these figures are being plotted.

Trev/Paul to check

It would be helpful to include in the manuscript a table (or something similar) where the required inputs, available options (and what happens when the user selects each option), and outputs of each uncertainty method are at least listed (even better if explanations of the generated outputs are provided).

To be considered – Trev/Paul to check

**5. Treatment of autocorrelation**

I am not sure if the autocorrelation implementation is correct. Any references? I would suggest checking Evin et al. (2013, 2014) and Vrugt et al. (2022).

To be checked – Paul & Trev

G. Evin, D. Kavetski, M. Thyer, and G. Kuczera. Pitfalls and improvements in the joint inference of heteroscedasticity and autocorrelation in hydrological model calibration. Water Resources Research, 49(7):4518–4524, 2013.

G. Evin, M. Thyer, D. Kavetski, D. McInerney, and G. Kuczera. Comparison of joint versus postprocessor approaches for hydrological uncertainty estimation accounting for error autocorrelation and heteroscedasticity. Water Resources Research, 50(3):2350–2375, 2014.

Vrugt, J. A., de Oliveira, D. Y., Schoups, G. & Diks, C. G. On the use of distribution-adaptive likelihood functions: Generalized and universal likelihood functions, scoring rules and multi-criteria ranking. Journal of Hydrology 615, 128542. doi:10.1016/j.jhydrol.2022.128542 (2022).

**6. Limitations/advantages of each method and additional guidelines**

Some guidance could be provided for selecting from the different uncertainty estimation methods available. A summary of the limitations/advantages of each method would be helpful, as well as guidelines for the specification of the required parameters (for example, studies like McInerney et al., 2017 could be provided as a reference for helping users to select initial Box-Cox parameter values).

This will be included – KB what is each method doing differently

Box-Cox – check past papers about fitting.  Different folder with fitting algorithm not working?

Negative uncertainties

McInerney, D., Thyer, M., Kavetski, D., Lerat, J., and Kuczera, G. (2017), Improving probabilistic prediction of daily streamflow by identifying Pareto optimal approaches for modeling heteroscedastic residual errors, Water Resour. Res., 53, 2199– 2239, doi:10.1002/2016WR019168.

**7. Going beyond recording assumptions**

The toolbox includes a condition tree that users can use to document choices and assumptions. It would beneficial if the toolbox also included the option of testing (some of) these assumptions. For example, if the uncertainty analysis is conducted using a formal likelihood/Bayesian inference, common checks would be to evaluate the residual assumptions and the quality of the uncertainty estimates (see Thyer et al., 2009 and many others).

M. Thyer, B. Renard, D. Kavetski, G. Kuczera, S. W. Franks, and S. Srikanthan. Critical evaluation of parameter consistency and predictive uncertainty in hydrological modeling: A case study using Bayesian total error analysis. Water Resources Research, 45(12), 2009.

While this might be beyond the scope of the paper, at least some discussion on how the assumptions can be tested could be included, especially for "informal" metrics (given that in some cases the assumptions are not known, how can we evaluate the reliability of the results?).

The referee knows, of course, that this is really only possible for the assumptions of formal statistical methods, for which some tests are available and demonstrated.  The sources of epistemic uncertainty mean, however, that there can be no right answer only assumptions that seem reasonable in the particular application.  This is why we stress the associated audit trail, so that users and those that come later can review the audit trail and see if they find the assumptions acceptable.

**Minor comments**

1.  What is the difference between the GLUE-Limits of Acceptability (LoA) method and the use of LoA within the DREAM toolbox (Vrugt & Beven, 2018)?

This is now discussed

J. A. Vrugt and K. J. Beven. Embracing equifinality with efficiency: Limits of acceptability sampling using the DREAM(LOA) algorithm. Journal of Hydrology, 559:954–971, 2018

2. McInerney et al. (2018) could be added to the introduction.

This has been done.

D. McInerney, M. Thyer, D. Kavetski, B. Bennett, J. Lerat, M. Gibbs, and G. Kuczera. A simplified approach to produce probabilistic hydrological model predictions. Environmental Modelling Software, 109:306–314, 2018.

3. Residuals and errors are used interchangeably, but they do not have the same meaning. From Vrugt & de Oliveira (2022): "The word error implies a difference between an observed value and its true value. As our measurements of system behavior are imperfect, the residuals are estimates of the errors under the assumed model. Hence, we should use the word residual instead."

This is correct.   The revision is much more careful in revising

Vrugt, J. A. & de Oliveira, D. Y. Confidence intervals of the Kling-Gupta efficiency. Journal of Hydrology 612, 127968. doi:10.1016/j.jhydrol.2022.127968 (2022).

4. Glossary: some definitions are somewhat informal and not necessarily accurate. Even though the toolbox was designed for individuals "who are not necessarily experts in uncertainty estimation", it is important to define the terms accurately.

The Glossary definitions have been reviewed and revised

5. Code, Workflow 1: "This Workflow is a CASE STUDY demonstrating forward uncertainty analysis for the CHASM landslide model (see CHASM_IO_files_2014.pdf in CHASM folder)": I couldn't find the "CHASM_IO_files_2014.pdf" file

This has been corrected

6. What is the difference between a "case study" (workflow) and "application example"/"example" (other workflows)?

Usage of case study / workflow / application has been reviewed throughout.  Case study does not now appear in the manuscript.   Applications have been linked to Worflows where possible,

7. Website, Learn about CURE and Uncertainty Estimation: Update link and reference to Page et al. (2021)

This has been modified

8. Website, Case Studies: Why only 4, 5, and 11 are presented here?

To be checked - can all be included.